# The Role of Online Social Support in Mental Health: Comparing Rural and Urban Youth [note 1]

**DOI:** 10.3390/children12020113

**Published:** 2025-01-21

**Authors:** Ellen A. Knowles, BreAnne A. Danzi

**Affiliations:** Department of Psychology, University of South Dakota, 414 East Clark Street, Vermillion, SD 57069, USA; breanne.danzi@usd.edu

**Keywords:** social media, online social support, adolescents, depression, anxiety, youth

## Abstract

Background/Objectives: Social support is essential to development, mental and emotional health, and reducing risk for psychological distress. Youth are exposed to various opportunities for socialization with peers, including through social media. Strong online social networks may be most critical for youth from isolated communities, including those from rural areas or LGBT+ (lesbian, gay, bisexual, transgender, etc.) youth. With this study, we investigated associations between social media use and online social support among adolescents. Additionally, we compared the online experiences of rural and urban youth. We also examined associations between online social support and psychological functioning and investigated whether the strength of these relationships was moderated by community type (rural versus urban). Finally, we evaluated the online experiences of LGBT+ youth. Methods: Two hundred seventy-five participants (ages 13–19) from seven high schools located in either rural or urban communities completed a paper and pencil survey on their mental health, social media use, and online social support. Results: Social media use was positively associated with online social support. Rural youth were found to report greater social media use and online social support compared to urban youth. Online social support was not associated with depression or anxiety; community type (rural versus urban) did not moderate the strength of these relationships. LGBT+ youth in rural communities reported greater social media use and support online compared to LGBT+ youth in urban communities. Conclusions: These findings shed light on the potential positive benefits of social media use as it is related to perceptions of social support among online friends. They demonstrate that interactions online may contribute to stronger support networks within adolescents, particularly among vulnerable youth. Findings suggest social media may increase accessibility to social support for youth who may be at risk for social isolation.

## 1. Introduction

### 1.1. Social Support

Support from peers and family has been identified as a major factor in promoting healthy development for youth globally [1] by supporting mental and emotional development across the lifespan [2,3,4]. Extant research has demonstrated that social support plays an important interdisciplinary role in enhancing health and well-being [5] and has been found to reduce the risk for psychopathology, such as depression and anxiety [2,3,4,6]. However, it may be that not all types of social support are equally beneficial. For instance, Cavanaugh and Buehler [7] found that parental and peer support were uniquely associated with reduced risk for different aspects of psychological well-being (i.e., loneliness and social anxiety, respectively).

Additionally, social support may not provide the same degree of benefits across individuals. That is, the broad benefits of social support have been found to be particularly robust among youth from diverse identity groups, such as youth from the LGBT+ community [8,9]. Conversely, poor social support may have especially detrimental effects on the psychological well-being of vulnerable groups [10]. Importantly, adolescence marks a key developmental stage characterized by a great deal of change in one’s social–emotional functioning, making it increasingly important for teens to feel connected to, and accepted by, their peers [11]. Thus, identifying how to amplify social support among young populations is critical to promoting healthy psychosocial functioning for youth, particularly those from diverse or underserved backgrounds.

### 1.2. Social Support Among Rural Youth

Previous studies have found that the mental, physical, and overall well-being of youth from rural areas is heavily related to social support from peers, family members, and other community members [12]. This indicates that, when present, stronger perceptions of social support may be exceptionally beneficial for rural youth. In contrast, several studies have found that poor social support may have particularly detrimental effects on the psychological well-being of vulnerable groups [4,13]; this includes rural youth who may experience limitations to social connection due to their geographical isolation [14,15], making the need for meaningful interpersonal connection even more urgent.

Although these youth tend to report strong feelings of connectedness with their community, the subset of rural youth who feel socially isolated may experience even greater risk for detriments to their mental health, such as loneliness and boredom [15]. For these youths, this dichotomy of emphasized importance of social support and poor access to such support may exacerbate feelings of isolation and related mental and emotional difficulties.

### 1.3. Psychosocial Vulnerability of Rural Youth

Adolescents’ environments can play a critical role in influencing their mental and emotional development [16], and exposure to adversity during this period can have lasting impacts [17]. Youth from a range of community types are vulnerable to various risk factors that may contribute to early life stress and subsequent psychopathology [18]. Urban youth, for example, are known to be at increased risk for exposure to traumatic events and community violence [19]. However, they are also more likely to live closer to healthcare facilities and live above the poverty line [20]. Alternatively, rural youth are at greater risk for substance use, parent unemployment, and poverty [21]. These factors may play a dynamic role in perpetuating negative community perceptions of mental health conditions and care [14] and may increase the risk of impacted psychosocial functioning [22]. Additionally, rural youth tend to experience greater barriers to accessing health care [20] and have been found to encounter greater difficulty developing a sense of trust in the health system overall [23]. These limitations contribute to increased stigma against health care, particularly mental health care [14], which can lead to delayed intervention and, subsequently, worse psychosocial functioning [24].

Additionally, LGBT+ youth from rural communities are at greater risk for being subject to peer victimization and harassment, which has been associated with low perceptions of social support and other negative outcomes, including increased psychopathology and engagement in risky health behaviors such as substance use [8,9,25]. While being a diverse individual in a rural community does not necessitate social isolation, research indicates that it may contribute to a reduced sense of belongingness and increased risk for psychopathology [9,25]. Thus, identifying ways to bolster social support among these youth is critical to protecting against negative outcomes.

### 1.4. Role of Social Media

Since the rapid rise of technology, the frequency and quality of interpersonal interactions among youth have expanded exponentially. Today, social media is a widely used tool for facilitating communication and socialization among youth [26]. However, despite its widespread utilization, findings on how social media use is associated with mental health and well-being are less than clear.

Much of the early literature on social media use demonstrated support for the negative outcomes associated with social networking, such as depression and anxiety [27,28]. This research often focused on examining the relationships between negative outcomes and problematic social media use, which was characterized by constructs such as social media addiction and excessive use (e.g., [29]). Among youth, concerns associated with social media use include increased risk for peer victimization through cyberbullying as well as exacerbated body image issues and related disorder eating behaviors [30]. Furthermore, social media use and its associated outcomes likely differ across youth. For instance, youth with pre-existing mental health conditions may be at risk for spending more time on social media compared to their peers with no mental health diagnoses [31]. Excessive use has been found to be correlated with higher depression, anxiety, and stress in young people [32]. Therefore, the most psychologically vulnerable youth may experience a heightened risk of engaging in problematic social media use, which may aggravate their existing mental distress. However, despite the evidence for negative consequences of social media use, there appears to be a high degree of variability across studies in methodology, design, and participant characteristics [33], making it difficult to draw definitive conclusions regarding these relationships.

As such, more recent work has begun to challenge the belief that social media use is only associated with negative outcomes and has emphasized the importance of assessing possible benefits as well. For instance, a recent umbrella review of adolescent social media use and mental health found support for possible positive outcomes of social media use, including improved general well-being, happiness, and life satisfaction [34]. Additional benefits identified among adolescent populations specifically include increased access to social communication, learning opportunities, and identity exploration, which are all considered important components of adolescent development [30,35].

However, to further complicate the picture, other studies have found no relationship, positive or negative, between social media use and well-being. For example, an 8-year longitudinal study by Coyne and colleagues [36] evaluated 500 individuals from adolescence through emerging adulthood. They found that increased social media use within individuals was not associated with worse depression or anxiety across time, nor was decreased use associated with improved mental health. These mixed results across studies suggest that this topic warrants further investigation; in particular, it is important for research to evaluate both the positive and negative outcomes associated with social media use.

### 1.5. Social Media Use and Social Support

Within the past decade, online social support has emerged as a possible benefit of using social media. Some studies have suggested that social media may provide a valuable source of social support for various communities, particularly when used for the purpose of engaging with others [30,35,37,38,39,40]. In adults, studies have demonstrated that one’s number of online connections may be associated with greater perceived social support and, thus, improved psychological well-being, particularly in isolated populations and among individuals with similar life experiences [40,41,42]. In youth, social media has been found to provide an additional pathway through which youth can connect with one another and build a digital support system [30,35,43]. Additionally, social media use may increase access to social connections with others outside of one’s immediate community as well as contribute to improved abilities to connect with existing friends [40,41,42,44].

Furthermore, youth may seek specific types of social support online (e.g., informational, emotional) by using different platforms depending on their intended outcome [45]. There is some evidence that platforms offer unique benefits and that, like in-person support [7], not all platforms offer the same type of social support. For example, video games may promote enhanced well-being and strengthen intergroup relations [46], while platforms like Facebook support relationship building and the expansion of interpersonal connections [40]. Additionally, sites like Twitter and Reddit may be best used for seeking information on topics of interest and sharing experiences with others [45]. While these findings suggest that different types of social media offer unique benefits, few studies have directly investigated associations between individual social media platforms and different subtypes of online social support. This is an important next step to expanding the literature base, as it will aid in clarifying how and when youth benefit from their social media use. Additionally, uncovering what individual factors contribute to positive and negative outcomes of social media use across platforms is essential to understanding these relationships in greater depth.

### 1.6. Social Media Use Among Rural Youth

Little research has investigated the utility of social media use among adolescents from rural communities. This is an important area of investigation as rural youth generally are at increased risk for social isolation [15] and are also more likely to be frequent social media users [47]. Several studies have evaluated the effects of social media use among rural adolescents globally. Most of these investigated only the negative outcomes of social media use in rural youth; these include one study of adolescents and emerging adults in Australia [48], two on the psychological impact of social media use among rural Indian youth [49,50], two on social media addiction among rural students in Indonesia [51] and rural youth in India [52], and one on associations between social media use and tobacco use among adolescent boys in the rural US [53].

In addition to investigating the negative effects, a few have also evaluated the positive effects of social media use among rural populations. These include one study of youth in rural Pakistan [54] and another in rural India [55], both of which evaluated positive and negative associations with social media use. Additionally, two US studies investigated social media use among rural youth and demonstrated evidence for positive associations between technology use and community connectedness [56,57].

Thus, while there has been some investigation into how rural youth are affected by social media, there are few studies that have assessed both positive and negative outcomes associated with social media use. Furthermore, online social support appears to be understudied among rural adolescents. Given the ways in which rural youth benefit from in-person support [12], it is important to identify whether online social support might function in similar ways. This knowledge would add depth to the evidence base on how social media use can augment in-person social support and strengthen the support networks of vulnerable and isolated youth.

Additionally, as some rural youth are at even greater risk for social isolation [15], it is important to understand how social media use and related outcomes differ for these adolescents. For instance, LGBT+ rural youth represent a subset of already small communities. This may make interpersonal connections more difficult [15], thereby increasing the need for alternative avenues for social interaction. Clarifying the role of social media among these youth is a useful step in identifying how rural LGBT+ adolescents interact with others outside of their immediate communities and strengthen their support networks [45,58].

### 1.7. Social Media Use Among LGBT+ Youth

While research on social media use in this population is limited, there is some early evidence for the psychosocial benefits of social media use among LGBT+ youth. Several studies found that socially isolated LGBT+ youth in rural communities used social media for a variety of reasons, including seeking informational and emotional support, as well as interacting with others with shared experiences [45,58]. Another study of sexual and gender minority youth combined rural and other non-metropolitan youth and found that these youth used social media for various types of social support, including establishing relationships, seeking information, and self-expression [59]. Additionally, transgender youth have been found to benefit from using social media to obtain social and emotional support from peers [41]. LGBT+ youth may also encounter greater concerns about personal safety in school or other public settings, which may make online connection more desirable [60]. These youth are also more likely than their non-LGBT+ peers to report that their friends online were better able to provide support than their friends in person [60].

## 2. Current Study

Little is known regarding how rural youth compare to urban youth in terms of their social media use and associated outcomes. This is the first study to our knowledge to compare these associations among youth from rural versus urban communities. The present study assessed four primary research questions regarding the relationships between social media use, online social support, and psychological well-being in rural and urban youth.

**Aim** **1:**
*Determine whether social media use is associated with increased perceived online social support in youth.*


**H1:** 
*Greater frequency of social media use will be associated with higher perceived online social support in youth.*


**Aim** **2:**
*Compare social media use and perceived online social support between rural and urban youth.*


**H2a:** 
*Rural youth will report greater social media use compared to urban youth.*


**H2b:** 
*Rural youth will report greater perceived online social support compared to urban youth.*


**Aim** **3:**
*Evaluate the relationship between perceived online social support and depression and anxiety symptom severity in rural versus urban youth.*


**H3a:** *Community type (rural *vs. *urban) will moderate the relationship between perceived online social support and depression such that there will be a negative association between online social support and depression, and this relationship will be stronger for rural youth.*

**H3b:** *Community type (rural *vs.* urban) will moderate the relationship between perceived online social support and anxiety such that there will be a negative association between online social support and anxiety, and this relationship will be stronger for rural youth.*

**Aim** **4:**
*Examine social media use and online social support in LGBT+ youth.*


**H4a:** 
*Social media use will be associated with online social support among LGBT+ youth from rural and urban communities.*


**H4b:** 
*Social media use and online social support will be higher in LGBT+ youth from rural communities compared to LGBT+ youth from urban communities.*


## 3. Materials and Methods

### 3.1. Participants

Participants were 275 adolescents recruited from seven high schools (five public, two private) with student body sizes ranging from 51 to 1478; 45% (*n* = 123) of participants were recruited from rural schools, and 55% (*n* = 152) were recruited from urban schools. Participants were between the ages of 13 and 19 (M = 16.39; SD = 1.33). The participants were 66% biologically female and 6% gender diverse. Additionally, regarding sexual orientation, 79% reported they were straight. The participants were 8% Hispanic/Latine and 84% White. Additional demographic information is provided in Table 1.

### 3.2. Procedure

All study procedures were approved by the University of South Dakota Institutional Review Board (IRB-22-23 approved 28 June 2022). Participants were eligible for the study if they were (1) youth aged 11 to 19; (2) attended a school in a rural or urban county; (3) had informed consent from a parent or guardian to participate in the study; and (4) could complete the questionnaire in English. If youth were above the legal age of majority, they were able to provide their own informed consent for participation. Only schools located in areas classified as rural or urban were contacted for participation. These classifications were determined using the definitions provided by the Office of Management and Budget [61], in which “urban” is defined as over 50,000 people and “rural” is defined as less than 10,000 people. Schools in “micropolitan” areas (10,000–50,000 people) were excluded from this study.

Study personnel attempted to contact all rural and urban schools within a two-hour radius across three states to request permission to present the study to their student body. This search was eventually expanded to include schools within a five-hour radius. Approximately 80 schools were contacted, and of these, seven agreed to participate. Schools from rural and urban counties declined to participate for a range of reasons, including being in transitional periods, having limited time to dedicate to research, and school or district policy restrictions that did not allow research to be conducted.

Participant recruitment at participating schools was conducted over the course of nine months. Schools were visited approximately five times on average to distribute and collect consent forms and to collect data, with visit frequencies ranging from two visits to more than fourteen visits for each school.

At school visits, study personnel spoke to eligible students in their classrooms, auditoriums, and/or cafeterias (the method varied based on the preferences of each school principal). Students were made aware of the study’s purpose and procedures, including any potential risks and benefits associated with their participation. They were informed that at least some social media and/or phone use was required in order to participate in the study. They were then provided with a flyer and consent form to bring home to their caregiver. Students were informed that they would need to return a signed consent form on or before the day of data collection if they wished to participate in the study. If consent forms were returned prior to the day of data collection, they were stored in a secure location, and their names were recorded so study personnel could confirm their eligibility at the time of data collection.

On the day of data collection, students were gathered in a private space to complete the survey, per school preferences. Students with signed consent forms were then presented with an assent form if they were below the age of legal majority and were asked to provide their written assent if they wished to continue. Participants then completed a brief paper and pencil survey. After completing the survey, participants were given a debriefing form, which contained local and national mental health resources as a safeguard in case any questions caused the participant distress. The entire survey took approximately 20–30 min to complete. Participants received a $10 Amazon gift card as compensation following the completion of the survey.

### 3.3. Measures

The questionnaire included an evaluation of demographic information as well as measures that assessed social media use frequency, perceived online social support, anxiety symptoms, and depression symptoms.

### 3.4. Online Social Support and Social Media Use Frequency

Perceived online social support and frequency of social media use were assessed using the Online Social Support Scale (OSSS) [62]. This established self-report measure includes two parts. The first part asks respondents to indicate the frequency with which they use a range of social media platforms to connect or interact with other individuals (e.g., Snapchat, YouTube, texting). Respondents reported their use frequency for each platform individually. Responses are rated on a 5-point scale (0 = Never, 1 = Rarely, 2 = Sometimes, 3 = Pretty Often, 4 = A Lot). Respondents were then instructed to think about the online spaces they use and rate the frequency with which they experienced certain situations while interacting with others online over the previous two months. This portion of the measure includes 40 items, and responses are rated on the same 5-point scale as the first part of the measure (0 = Never, 1 = Rarely, 2 = Sometimes, 3 = Pretty Often, 4 = A Lot). The OSSS provides a total summary score (sum of all items; 1–40) in addition to four subscale scores: esteem/emotional support (sum of items 1–10), social companionship (sum of items 11–20), informational support (sum of items 21–30), and instrumental support (sum of items 31–40). Internal consistency for the overall scale in the present study was excellent (α = 0.97).

### 3.5. Anxiety

Anxiety symptoms were measured using the Generalized Anxiety Disorder Screener (GAD-7) [63], a widely used 7-item self-report questionnaire that asks respondents to rate the frequency with which they have been bothered by various symptoms of anxiety over the past two weeks. Responses were reported on a 4-point scale (0 = Not at all, 1 = Several days, 2 = More than half the days, 3 = Nearly every day). Total scores were calculated by summing all the items, resulting in a possible score range of 0 to 21. Additionally, respondents were asked a final question about how difficult the issues made their day-to-day lives, which they ranked on a separate 4-point scale (0 = Not at all, 1 = Somewhat difficult, 2 = Very difficult, 3 = Extremely difficult). Internal consistency for the present study was excellent (α = 0.90).

### 3.6. Depression

Symptoms of depression were assessed using the Center for Epidemiological Studies for Depression scale (CES-D) [64], a 20-item self-report measure originally developed as a screener for clinical depression [65,66]. Respondents were asked about their feelings over the past week and rated their responses on a 4-point scale (0 = Rarely or none of the time (less than 1 day), 1 = Some or a little of the time (1–2 days), 2= Occasionally or a moderate amount of time (3–4 days), 3 = Most or all of the time (5–7 days)) with even items scored in reverse. A total score is found by summing all items, resulting in possible scores ranging from zero to 60, with higher scores denoting the presence of greater symptomatology. Internal consistency for the present study was excellent (α = 0.94).

### 3.7. Statistical Analyses

Statistical analyses were conducted using the Statistical Package for the Social Sciences (SPSS v.29). Two hundred eighty-three participants were recruited from seven participating high schools. Eight students were excluded due to missing one or more entire measure(s), resulting in 275 total participants. The frequency of missing data for the primary observed variables was 0.003%. Missing data were handled using mean imputation. Given the very small amount of missing data presently, single imputation was an appropriate method [67]. Prior to analyses, the data were cleaned and checked for outliers, normality, and distribution of variables. The skewness and kurtosis of all primary observed variables were within acceptable ranges [68]. However, some evidence of heteroscedasticity was detected among depression (CESD) scores. To account for this, a weighted least squares regression moderation model was used for Hypothesis 3a.

Independent samples t-statistics and chi-square statistics were used to evaluate group equivalence (Table 1). For Hypothesis 1, a Pearson’s correlation was used to evaluate the presence of an association between social media use and perceived online social support in youth. To control for differences between the demographic makeup of the rural and urban participants, partial correlations were also conducted (see Table 2). For Hypothesis 2a, a one-way Analysis of Covariance (ANCOVA) was used to compare differences in social media use patterns and behaviors between rural and urban youth after controlling for race/ethnicity, age, and financial challenges. For several social media platforms (Facebook, chatting services, Twitter, Yik Yak, Other), the Levene’s test was significant; for these, a Quade Nonparametric ANCOVA was used instead. For Hypothesis 2b, a one-way ANCOVA was used to assess and compare perceived online social support between rural and urban youth after controlling for race/ethnicity, age, and financial challenges. Hypotheses 3a and 3b required linear regressions with interactions to determine the presence of an interaction effect between community type and online social support on depression and anxiety. Specifically, these analyses were used to identify whether community type (rural or urban) moderated the strength of the relationship between perceived online social support and psychopathological outcomes. Again, race/ethnicity, age, and financial challenges were all included in the models as covariates. Finally, for Hypothesis 4, a one-way ANCOVA was used to evaluate the association between social media use and online social support among LGBT+ youth after controlling for race/ethnicity, age, and financial challenges. All analyses evaluating online social support were also repeated for each subscale of online social support.

## 4. Results

### 4.1. Descriptive Statistics

Descriptive statistics, including means, standard deviations, and bivariate correlations between all primary observed variables, are reported in Table 2. Partial correlations were completed to identify associations between study variables after controlling for participant race/ethnicity, age, and financial challenges. These variables were not observed to confound any associations between the primary variables of interest.

### 4.2. Rural Versus Urban Group Differences

The urban and rural participants were found to differ on several study variables. Although sex, gender identity, and sexual orientation were similar between groups, the rural participants were more diverse in both race and ethnicity. Specifically, 14% of rural participants were reported to be of Hispanic/Latine ethnicity, while only 2% of urban participants reported being of this ethnic background. Additionally, 23% of rural youth reported belonging to at least one non-White race, while only 11% of urban participants were racially diverse. Additionally, the rural participants were significantly younger than the urban participants. On average, urban youth were around 16–17 years of age (M = 16.71, SD = 1.24), while rural youth were 15–16 years old (M = 15.98, SD = 1.33). Finally, the participants differed economically as 27% of rural participants reported needing free or reduced lunch (referred to as financial challenges below), while 8% of urban participants reported needing this financial support. All demographic information is provided in Table 1.

### 4.3. Primary Observed Variables

On average, participants scored just below the clinical cutoff for depression on the CESD. However, 38% scored above the clinical cutoff, which suggests these youth may be at increased risk for a depressive disorder [65]. Additionally, average scores for anxiety fell in the moderate range, with 44% of all participants scoring above the clinical cutoff, suggesting heightened risk for an anxiety disorder [69]. When comparing identity groups, youth from the LGBT+ community were found to report significantly higher depression (M = 24.78, SD = 13.33) and anxiety (M = 10.56, SD = 5.21) compared to their heterosexual cis-gender peers’ depression (M = 12.63, SD = 11.73) and anxiety (M = 6.26, SD = 5.06).

Average scores for total online social support among all participants were in the middle of the measure, suggesting moderate levels of perceived online social support. Total scores for the four subscales of online social support ranged from 0 to 40 across all subscales. Paired samples *t*-tests were used to investigate whether average scores among online social support subscales differed. These revealed that average scores for instrumental support were significantly lower than those of esteem/emotional support (t(273) = 15.77, *p* < 0.001), social companionship (t(273) = 17.50, *p* < 0.001), and informational support (t(273) = 19.97, *p* < 0.001). Average scores between other scales were not found to differ significantly (all *p*s > 0.05).

### 4.4. Associations Between Social Media Use and Online Social Support

A partial correlation revealed a weak positive association between social media use and online social support after controlling for all covariates. We also examined the bivariate correlations between online social support and social media platform use, as well as the partial correlations controlling for race/ethnicity, age, and financial challenges. These covariates were not found to have confounding effects on the correlations; thus, only the bivariate correlations are reported below. Online social support was found to be weakly positively associated with total social media use as well as the use of several platforms (Table 3). Total social media use was also positively associated with all subscales of online social support.

### 4.5. Social Media Use in Rural Versus Urban Youth

Overall, adolescents reported using a wide range of online social networking platforms (Figure 1). The top three most frequently used social networking platforms were texting, Snapchat, and TikTok.

When evaluating the differential use of each platform, rural youth and urban youth were not found to differ in the frequency with which they used most platforms (See Figure 2). However, rural youth did report using Snapchat, Facebook, and Yik Yak significantly more frequently than their urban peers. However, these effects were small (all ηp2’s = 0.03), indicating community type had a small, albeit significant effect on platform use.

### 4.6. Online Social Support in Rural Versus Urban Youth

Rural youth reported significantly higher perceived online social support overall compared to urban youth on average (see Table 4), although this effect was small. This was also true across all four subscales of social support. The effect sizes of community type on all subscales of online social support were small (see Table 4).

### 4.7. Social Support and Mental Health by Community Type

In the first moderation model, online social support did not have a significant main effect on depressive symptoms, indicating that depressive symptom severity did not differ for youth depending on the level of their online social support perceptions. However, there was a significant main effect for community type on depressive symptom severity, such that depressive symptom severity was higher in urban youth compared to their rural counterparts. There was not a significant moderating effect of community type on the association between online social support and depression. All results are reported in Table 5. This model was repeated with each subscale of online social support and the same findings remained true across all four subscales. That is, depressive symptom severity remained higher among urban youth (all *p*s < 0.05), but no other main effects or moderating effects were found to be significant (all *p*s > 0.05).

The second model demonstrated there was not a significant main effect of online social support on anxiety symptom severity, indicating that anxiety symptom severity did not differ for youth depending on the level of their online social support. However, there was a significant main effect for community type on anxiety symptom severity, such that anxiety symptom severity was higher for urban youth than it was for rural youth. Similarly to the first model, there was not a significant moderating effect of community type on the association between online social support and anxiety. All results are presented in Table 5. Again, this model was repeated using each subscale of online social support, and the same findings remained true across all four subscales. That is, anxiety symptom severity was higher among urban youth (all *p*s < 0.05), but no other main effects or moderating effects were found to be significant (all *p*s > 0.05).

### 4.8. Social Media Use and Online Social Support Among LGBT+ Youth

Online social support and social media use were moderately correlated with one another among LGBT+ youth (r = 0.43, *p* < 0.001). Overall, social media use frequency did not differ between LGBT+ youth and their non-LGBT+ peers (*p* > 0.05). However, LGBT+ youth reported significantly greater use of “other” platforms compared to their heterosexual cis-gender peers (*p* < 0.001). Furthermore, within LGBT+ youth alone, those from rural communities reported greater social media use overall (see Table 6). Frequency of using Snapchat (F (1, 57) = 4.44, *p* = 0.040), chat services (F (1, 57) = 12.89, *p* < 0.001), and “other” platforms (F (1, 57) = 12.35, *p* < 0.001) were significantly higher among rural youth compared to urban youth. These were the only platforms in which use frequency differed between rural and urban youth.

Overall, online social support was not found to differ between LGBT+ youth and their non-LGBT+ peers. This was also true for all four subscales of online social support. However, when comparing rural and urban youth within the LGBT+ community, rural youth were found to report greater perceptions of online social support compared to urban youth. This was also true for esteem/emotional support, social companionship, and instrumental support, but not informational support. All results are reported in Table 6.

Among LGBT+ youth, online social support was only found to be weakly associated with Facebook use (r = 0.27, *p* = 0.038) and moderately associated with the use of chat services (r = 0.35, *p* = 0.008). Overall, online social support was not significantly associated with the use of any other platforms (all *p*s > 0.05). Social media use broadly was weakly associated with informational support (r = 0.28, *p* = 0.034) and instrumental support (r = 0.28, *p* = 0.030), whereas moderate associations were found between social media use and esteem/emotional support (r = 0.48, *p* < 0.001), as well as social companionship (r = 0.47, *p* < 0.001).

Finally, among LGBT+ youth, neither online social support (β = 0.05, SE = 0.08, *p* = 0.505) nor community type (β = −2.69, SE = 4.33, *p* = 0.537) had significant main effects on depressive symptom severity. Additionally, consistent with what was found among all participants, community type did not moderate the strength of the association between online social support and depression (β = −0.04, SE = 0.11, *p* = 0.744). Additionally, neither online social support (β = 0.05, SE = 0.03, *p* = 0.076) nor community type (β = −2.78, SE = 1.75, *p* = 0.118) had significant main effects on anxiety symptom severity. Community type also did not moderate the strength of the association between online social support and anxiety (β = −0.02, SE = 0.04, *p* = 0.679). These models were repeated with each subscale of online social support, and the same findings remained true across esteem/emotional support and social companionship in both models, as well as for informational support in the depression model. That is, no main effects or moderating effects were found to be significant among LGBT+ youth (all *p*s > 0.05). However, informational support was found to have a significant positive main effect on anxiety (β = 0.376, SE = 0.089, *p* = 0.035) and instrumental support was found to have a significant positive main effect on both depression (β = 0.409, SE = 0.301, *p* = 0.046) and anxiety (β = 0.453, SE = 0.11, *p* = 0.023) such that these types of support were higher among youth with greater symptom severities.

## 5. Discussion

Social support has been identified as a key factor in bolstering well-being among youth, including buffering against negative consequences of early life stress [70]. The benefits of social support may be particularly important for youth from isolated communities, such as rural youth or youth from the LGBT+ community. With the present study, we investigated the role of social media in bolstering online social support for adolescents from rural and urban communities. We also assessed whether social media use and online social support were associated with reduced risk for depression and anxiety in adolescents from different community types. Finally, we evaluated the online experiences of rural and urban youth who belong to the LGBT+ community.

### 5.1. Associations Between Social Media Use and Online Social Support

Consistent with Hypothesis 1, greater social media use frequency was found to be broadly correlated with higher perceptions of online social support overall. These findings are in line with other studies [35,40,45,71], which have demonstrated the possible positive associations between social media use and perceptions of social support and belongingness. The present findings are not consistent with previous research, which focused on the potentially deleterious effects of social media use [27,28], including suggestions that social media has a negative impact on peer socialization and interpersonal functioning among youth [72]. The present study expands upon existing literature [34] by providing some indication that social media use may be associated with higher perceptions of social support among some youth. This study is also among the first to investigate various subtypes of online social support among youth, which provides further information about how social media use specifically benefits young users. Furthermore, while previous studies have identified the benefits of social media use among populations with highly unique life experiences [41,42,58], the present study builds upon these findings by comparing the potential associations between social media use and perceptions of online social support among youth from rural and urban communities.

Across participants, social media platforms were found to vary in their associations with the different subtypes of online social support. Interestingly, among the top three most used platforms (Texting, Snapchat, TikTok), none were positively associated with all measures of online social support. In fact, texting, despite being the most frequently used form of social networking, was not found to be associated with any measure of perceived online social support. Previous studies have identified texting as functionally distinct from other forms of social media use [73]. Our findings may provide further evidence for this conceptualization and suggest that texting, while frequently used, is seen as a more traditional form of digital communication, perhaps akin to speaking in person or over the phone, whereas other platforms offer opportunities for engaging with friends in unique ways. In contrast, despite being less frequently used, chat services, Twitter, and “other” platforms were the only social media platforms found to be associated with overall online social support as well as all subscales of social support. Below, we discuss each subtype of online social support and their associations with social media platforms in greater depth.

### 5.2. Esteem/Emotional Support

In the current study, we found that esteem/emotional support was associated with engagement in creative outlets (i.e., Snapchat, TikTok, and Instagram) as well as relational media (i.e., chat services) and self-media (i.e., Facebook and Twitter). These findings are consistent with previous studies, which have found that engagement with social networking sites in general may be associated with increased perceptions of emotional support and social capita [45,74]. These findings may indicate that engaging in outlets that promote self-expression and creativity may contribute to an improved sense of self and psychosocial well-being among youth. Both of which are important components of healthy individual and interpersonal growth in youth; this is an important area for future research.

### 5.3. Social Companionship

Social companionship was associated with relationship-building platforms (i.e., chat services) and self-media (i.e., Twitter). Social companionship is not unlike esteem/emotional support in that the platforms associated with these types of support are often marked by direct, private interaction with others. There is some evidence to suggest that private communication online can lead to greater disinhibition, which results in increased self-disclosure when interacting with others online [75]. Thus, it may be that social companionship is more easily accessed through platforms that include components of privacy, which may enable youth to express themselves openly. Furthermore, social companionship was positively associated with engagement in video games and creative outlets (i.e., YouTube). This was in line with previous research, which demonstrated that engagement with video games can improve intergroup relations and bolster perceptions of online social support [46]. These types of platforms may also offer anonymity in a way that others do not. Anonymity has been identified as an important feature of some platforms and, like privacy, may be an attractive component of online communication for individuals who are not comfortable sharing personal information publicly [76].

### 5.4. Informational Support

Informational support was associated with relational platforms (i.e., chat services) and self-media (i.e., Twitter, Snapchat, Instagram). Like those associated with esteem/emotional support, all these platforms require some level of direct and private communication. This may indicate that adolescents use the same types of platforms to build relationships and exchange knowledge or information with friends. This is consistent with previous findings, which have demonstrated that young individuals are likely to use social media to share information with their peer groups, particularly during important life transitions such as adolescence [74]. It is possible that the platforms used for informational and emotional support overlap due to youth interacting with the same people for both purposes; however, additional research is needed to address this question directly.

### 5.5. Instrumental Support

Instrumental support, which refers to giving and receiving help from others online, was associated with creative platforms (e.g., Snapchat and TikTok), self-media (e.g., Instagram, Facebook, and Twitter), and relational platforms (e.g., chat services). These results are also similar to the associations found for esteem/emotional support. Given the overlap between instrumental support and esteem/emotional support, it may be that adolescents are most likely to exchange help with others who they feel have offered them emotional support and with whom they have already established a relationship, either online or in person. Previous studies have found that adolescents tend to use social media to communicate with pre-existing friends [77]. While we did not inquire about whether adolescents were interacting with friends or people whom they knew in person, future research should investigate the extent to which adolescents know their online social network.

The present findings indicated that, among youth, there may be few platforms that bolster social support broadly, but there are several platforms that provide at least some degree of social support. Future research should build upon these findings to further clarify who adolescents are interacting with online and what types of support they are obtaining from in-person friends and online-only friends.

To our knowledge, this is the first study to compare how the use of various platforms is differentially associated with the four subtypes of online social support in youth. The present findings suggest that while social media use does provide general social support benefits, the particular types of benefits may differ depending on which platforms adolescents frequently use.

### 5.6. Social Media Use Among Rural and Urban Youth

Consistent with Hypothesis 2a, youth from rural communities were found to report greater overall social media use, regardless of platform type, compared to youth from urban communities. This is consistent with previous studies which have identified rural youth as among those with the highest frequency of social media use [47]. In addition to rural youth reporting greater overall social media use, they also reported more frequent use of several platforms, including Snapchat and Facebook. These platforms may be more popular among rural youth as they offer opportunities for youth to communicate directly with peers, share information, and offer support. Facebook has previously been identified as a means through which individuals can connect with people they already know in the real world [78]. As rural communities are known to prioritize the maintenance of interpersonal relationships [12], and rural youth have been found to augment their relationships with social media [57], it is possible that rural youth in the present study use Facebook and Snapchat to stay connected to other members of their in-person community in ways that urban youth do not.

Overall, the present findings indicate that rural youth may use social media more frequently than their urban counterparts. They also expand on the existing literature by identifying specific platforms that were more frequently used by rural youth. The next steps will be to continue clarifying what underlying mechanisms contribute to higher frequencies of social media use in rural youth and who they are interacting with online.

### 5.7. Perceptions of Online Social Support Among Rural and Urban Youth

In addition to reporting greater social media use frequency, rural youth also reported greater online social support overall, consistent with expectations. As has been previously stated, the well-being of youth from rural areas has been found to be heavily reliant on support from their community [12]. The present findings may suggest that despite possible limitations in socializing with like-minded peers in person, rural youth may engage with social media in ways that are supplemental to their in-person interactions with others. Furthermore, rural youth were found to report greater perceptions of support across all four subscales of online social support. These findings are consistent with previous research which has demonstrated that isolated youth not only engage with social media more but also report feeling greater connectedness with their online community [41,42].

### 5.8. Psychopathology, Social Media Use, and Social Support

There is myriad research highlighting the negative effects of social media use on mental and emotional well-being [29,30,31,32,33]. In the current study, however, social media use was not found to be associated with anxiety or depression among adolescents from rural or urban community types. Additionally, we did not find community type to have a moderating effect on the strength of the associations between online social support and depression or anxiety. That is, being from a rural or urban community did not moderate the strength of the associations between social support and psychopathology.

This lack of an association between social media use and psychopathology is consistent with previous research, which has found no relationship between the two constructs [36]. This study adds to the mixed evidence for the long-debated link between social media use and psychological distress and may support the notion that social media is not universally harmful.

Importantly, the majority of previous research on social media use and well-being was conducted prior to the COVID-19 pandemic. Since the onset of the pandemic, social media use and online social connections have evolved substantially [79,80]. Thus, it may be that the present findings illuminate this shift in how youth are using social media and how they are impacted by their interactions with others online.

In contrast to what was predicted in Hypotheses 3a and 3b, online social support was also not found to be associated with depression or anxiety, positively or negatively. As social support has previously been identified as a protective factor for youth in defending against stress [81], we hypothesized that online social support would work similarly by decreasing depression and anxiety. However, while it is possible that online social support is beneficial under stress, depression and anxiety are fundamentally different from stress and represent more significant psychological distress [82]. Thus, an explanation for the lack of associations found presently may be that online social support is not as protective in the face of more significant psychopathology beyond typical daily stress.

Furthermore, research shows that youth with depression are susceptible to experiencing blunted responses to social acceptance as well as lower cognitive flexibility [83], meaning individuals with depression are more likely to expect rejection and feel less accepted by peers. These perceptual distortions are likely to impact how these youth interpret their interactions online. Therefore, it may be that for youth with moderate to high levels of depression or anxiety, the distress associated with online interactions neutralizes the benefits of online social support.

### 5.9. Online Experiences of Rural and Urban LGBT+ Youth

LGBT+ youth from rural communities were found to report greater social media use and online social support compared to their urban LGBT+ peers, consistent with Hypothesis 4b. Additionally, rural LGBT+ youth endorsed higher perceptions of online social support across three of the four subscales compared to urban LGBT+ youth. These findings are consistent with previous studies that have identified social media as an accessible tool for engaging in social connection [43], particularly for youth from isolated communities [40,41,42]. Additionally, they build upon previous findings, which have demonstrated that LGBT+ youth from rural communities benefit socially from engaging with others online [45,58].

More specifically, greater use of Facebook and chat services was found to be associated with higher online social support among LGBT+ youth, whereas other platforms were not. As previously described, these types of platforms may support relationship building [78], which is a known priority for rural youth [12] as well as LGBT+ youth generally [41,60]. They also facilitate direct and private communication, which may be an important component of socialization, particularly among sexually- and gender-diverse youth who are at increased risk for harmful social interactions with peers [60]. Thus, these private platforms may offer a safe space for vulnerable youth to interact with friends. Furthermore, LGBT+ youth reported using unlisted platforms more frequently than their heterosexual and cis-gender peers. This finding may suggest that youth from this community engage frequently with platforms that are not mainstream or well-known by adults. As social media use has been identified as a popular tool used by the LGBT+ community to connect with peers [84], identifying these platforms is critical to clarifying how youth from these communities are interacting with and supporting others.

Finally, among LGBT+ youth, those who reported greater instrumental support also reported greater depression and anxiety. Instrumental support includes the provision of resources and needed services [62]. These results build upon previous studies of rural LGBT+ youth, which only discussed emotional and informational support [45,58]. These findings suggest that instrumental support may also be a potential benefit of social media use among this vulnerable subpopulation. More specifically, these findings indicate that LGBT+ youth with higher depression and anxiety may differ in how they engage with their online social network; however, further research is needed to better understand the relationship between these constructs.

### 5.10. Limitations and Future Directions

The present findings represent an important contribution to the field of child and adolescent psychology by providing insight into how social media use facilitates online social support among youth from diverse communities. The current study was methodologically strong in that it used paper-and-pencil data collection, allowing us to avoid biasing the participants towards youth who are more comfortable using online platforms in general, which is a relevant consideration given the nature of this study. This methodology also allowed us to abide by school policies and required a higher degree of regular contact with school personnel, which ultimately contributed to the maintenance of positive relationships between researchers and schools.

Nevertheless, several limitations should be noted. First, despite efforts to recruit a broad range of schools, only seven agreed to participate in the present study due to a variety of barriers. This may limit the generalizability of our findings as the participating states are not necessarily representative of all rural and urban schools across the nation. Additionally, due to schools being classified by county size, several small urban schools and relatively large rural schools were included, as were youth from a combination of both public and private schools. School size and school type may be factors of interest in future analyses, as this has been included in a previous study evaluating social support and psychopathology in youth [85]. Furthermore, it may be of interest to include the schools as possible covariates in future analyses to account for possible shared experiences among students from the same school. Future studies should investigate the various external factors that contribute to social media use, social support, and psychopathology among diverse student bodies.

Second, while the racial and ethnic make-up of the participants were largely representative of, if not more diverse than, the respective states from which they were recruited [86], the participants were majority White and non-Hispanic/Latine. Data were collected from multiple schools across states in an effort to recruit diverse and representative participants and increase the generalizability of the findings. While the present findings are believed to be generalizable to groups with similar demographic make-ups, it is unclear whether they are generalizable to more racially and ethnically diverse populations. Future research should continue to investigate these topics among diverse individuals and communities.

Third, while we included analyses of LGBT+ youth as we believe this to be an important population in need of further investigation, the number of participants in this subgroup was relatively small. Thus, the analyses conducted with this subgroup were underpowered and may have been susceptible to random variations, making the results non-generalizable to other LGBT+ groups. Regardless, few studies have compared LGBT+ youth to their non-LGBT+ peers, and even fewer have compared rural and urban LGBT+ youth to one another. Thus, while the present findings should be interpreted with caution, they also lay an important foundation for future work to build upon.

Finally, we did not ask youth to report the number of hours spent on their phones, texting, and on social media; rather, we used categorical items and asked respondents to select the option that best represented the frequency with which they used each platform. While frequency of social media use is an important construct, there are various methods that have been used to collect these data across studies [38]. The two most used self-report methods include asking respondents to report their time spent on social media and asking respondents to rate the frequency of their use, which is the method used presently [38]. While both methods are accessible to researchers, self-report measures do not always accurately capture the true frequency with which individuals engage with social media [87]. The gold-standard method for collecting use frequency data is often considered log-based measures [34]. However, this method requires more sophisticated software and resources [34], making it a less accessible tool. As log-based measures continue to evolve and accessibility improves, future research should attempt to employ both self-report and log-based methods to obtain more accurate data on actual use frequency and to compare these to participants’ perceptions of their use frequency.

## 6. Conclusions

The present study provided further insight into the ways in which youth from different communities use social media. For rural youth in particular, social media may offer an alternative pathway through which youth can build and maintain connections with peers and bolster their sense of social connectedness. Additionally, social media may provide an accessible means through which LGBT+ youth from rural communities can access peers and increase their social support network. Future research should continue to investigate what platforms are most preferred by youth from various communities and backgrounds, and for what reasons. Further exploration of these details will be an important component of understanding how to best promote safe and positive interactions with friends online among adolescents.

## Figures and Tables

**Figure 1 children-12-00113-f001:**
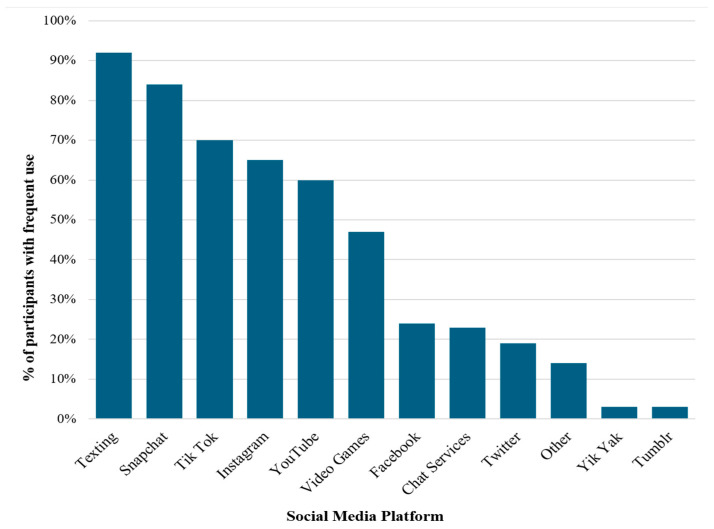
Social media platform use frequency. Note. Low frequency was defined as responses of “Never” or “Rarely”. High frequency is defined as responses of “Sometimes”, “Pretty often”, or “A lot”.

**Figure 2 children-12-00113-f002:**
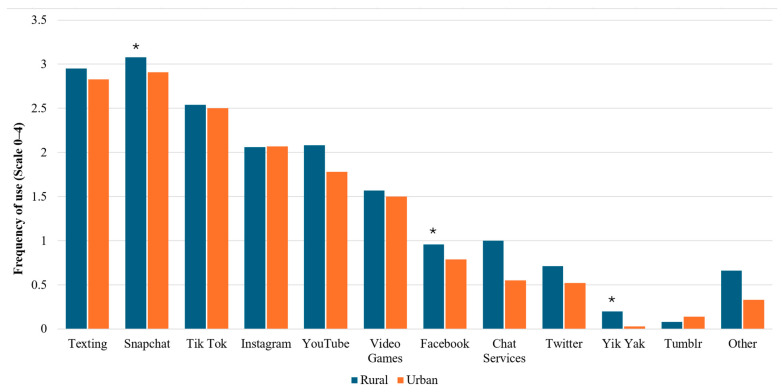
Differential use frequency of each social media platform between rural vs. urban youth. Note. * *p* < 0.01.

**Table 1 children-12-00113-t001:** Participant characteristics.

Variable	All Participants	Rural	Urban	Group Equivalence	*p*
N	275	123	152		
**Sex**				χ^2^(1) = 1.19	0.275
Female	66%	63%	69%		
Age *M *(*SD*)	16.39 (1.33)	15.98 (1.33)	16.71 (1.24)	*t*(273) = 4.67	<0.001
**Gender identity**				χ^2^(3) = 4.38	0.357
Girl	61%	55%	65%		
Boy	33%	36%	30%		
Transgender	1%	3%	1%		
Non-binary	3%	3%	3%		
Other	2%	3%	1%		
**Sexual Orientation**				χ^2^(3) = 3.07	0.381
Straight	79%	82%	77%		
Gay or Lesbian	2%	1%	3%		
Bisexual	10%	8%	12%		
Other	9%	9%	8%		
**Race**				χ^2^(6) = 14.90	0.021
White	84%	77%	88%		
Black	4%	6%	3%		
Asian	2%	2%	3%		
Pacific Islander	<1%	0%	1%		
American Indian/ Alaska Native	<1%	1%	0%		
Multiracial/Other	9%	14%	5%		
Ethnicity				χ^2^(1) = 14.19	<0.001
Hispanic/Latine	7%	14%	2%		
**Grade**				χ^2^(3) = 17.17	0.002
9th	16%	23%	11%		
10th	20%	22%	19%		
11th	30%	33%	27%		
12th	34%	22%	43%		
Free/reduced lunch	16%	27%	8%	χ^2^(1) = 17.81	<0.001

**Table 2 children-12-00113-t002:** Means, standard deviations, and bivariate correlations between primary observed variables.

Measure	M	SD	1	2	3	4	5	6	7
1. Depression	15.31	13.12	-						
2. Anxiety	7.24	5.42	0.77 **	-					
3. Social media use	16.96	6.02	0.01	0.04	-				
4. Online social support	68.06	33.59	0.03	0.04	0.37 **	-			
5. Esteem/emotional support	18.96	9.96	−0.01	0.05	0.33 **	0.84 **	-		
6. Social companionship	19.75	10.73	0.04	0.05	0.33 **	0.88 **	0.69 **	-	
7. Informational support	19.30	9.83	0.06	0.04	0.25 **	0.87 **	0.63 **	0.67 **	-
8. Instrumental support	10.05	8.96	0.03	0.01	0.32 **	0.80 **	0.52 **	0.58 **	0.67 **

** *p* < 0.001.

**Table 3 children-12-00113-t003:** Correlations between platform use and measures of online social support.

Platform	Social Support Type				
	Overall	Emotional	Companionship	Informational	Instrumental
Total social media use	0.37 **	0.34 **	0.33 **	0.25 **	0.32 **
Texting	0.04	0.06	0.04	0.02	0.01
Snapchat	0.20 **	0.30 **	0.10	0.12 *	0.16 **
Tik Tok	0.18 **	00.26 **	0.09	0.08	0.19 **
Instagram	0.19 **	0.27 **	0.10	0.14 *	0.15 *
YouTube	0.12 *	0.01	0.24 **	0.12	0.03
Video Games	0.10	−0.08	0.26**	0.08	0.07
Facebook	0.16 *	0.20 **	0.06	0.10	0.19 **
Chat Services	0.24 **	0.16 **	0.24 **	0.18 **	0.24 **
Twitter	0.26 **	0.25 **	0.24 **	0.18 **	0.20 **
Yik Yak	0.07	0.08	−0.02	0.09	0.11
Tumblr	0.001	−0.02	0.02	0.02	−0.02
Other	0.18 **	0.13 *	0.16 **	0.13 *	0.19 **

Note. * *p* < 0.05, ** *p* < 0.001.

**Table 4 children-12-00113-t004:** Social media use and online social support by community type.

Measure	Rural		Urban		F (1, 273)	*p*
	M	SD	M	SD		
Social media use frequency	18.07	6.56	15.97	5.38	7.26	0.007
Online social support	75.83	33.98	61.86	31.81	9.65	0.002
Emotional support	21.18	9.92	17.22	9.60	11.00	0.001
Social companionship	21.91	10.70	17.96	10.39	5.93	0.016
Informational support	21.15	10.06	17.87	9.38	6.83	0.012
Instrumental support	11.60	9.18	8.81	8.54	4.86	0.028

Note. M = mean; SD = standard deviation.

**Table 5 children-12-00113-t005:** Moderating effect of community type on online social support and psychopathology.

Predictor	B	SE	*p*	95% CI	
				RL	UL
Depression					
Online social support	−0.01	0.04	0.957	−0.07	0.07
Community type	−0.13	1.57	0.028	−6.55	−0.37
Online social support x Community type	0.08	0.05	0.374	−0.05	0.13
Anxiety					
Online social support	0.05	0.01	0.548	−0.02	0.04
Community type	−0.14	0.67	0.028	−2.80	−0.17
Online social support x Community type	0.01	0.02	0.877	−0.04	0.04

Note. B = regression coefficient; SE = standard error; CI = confidence interval. RL = Rural LGBT+; UL = Urban LGBT+. Community type was coded as 0 = urban, 1 = rural. Negative regression coefficients suggest that scores were higher for urban participants than for rural participants.

**Table 6 children-12-00113-t006:** Online social support and social media use among urban and rural LGBT+ youth.

Measure	Rural LGBT+		Urban LGBT+		F (1, 55)	*p*
	M	SD	M	SD		
Social media use frequency	19.48	6.06	14.74	4.74	11.56	0.001
Online social support	91.70	34.85	60.05	33.83	11.72	0.001
Emotional support	24.48	8.96	15.74	10.98	10.04	0.003
Social companionship	28.39	9.50	17.82	11.09	13.94	<0.001
Informational support	23.87	11.15	19.40	9.30	2.70	0.106
Instrumental support	14.96	9.48	7.10	7.30	12.48	<0.001

Note. M = mean; SD = standard deviation. *n* = 56.

## Data Availability

The data that support the findings of this study are available on request from the corresponding author (E.A.K.).

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
