# Peer review of "The Role of Online Social Support in Mental Health: Comparing Rural and Urban Youthâ€"

_children, 2025, doi:10.3390/children12020113_

Round 1
Reviewer 1 Report
Comments and Suggestions for Authors
Dear Authors,
Thank you for your submission of the manuscript titled "The role of online social support in mental health: Comparing rural and urban youth." Your study addresses an important and timely topic, exploring the relationship between social media use, online social support, and mental health outcomes among adolescents from rural and urban communities, while also investigating the experiences of LGBT+ youth. However, upon review, I identified several conceptual and methodological concerns that I believe could be addressed to strengthen the impact and clarity of your work.
- Challenges to the Study’s Framework
Your hypothesis that a higher frequency of social media use leads to greater online social support, and consequently to reduced depression and anxiety symptoms, presents an interesting starting point. However, the results of your study suggest that this hypothesis does not hold, as no significant associations were found between online social support and symptoms of depression or anxiety. This challenges the initial premise of your work.
A deeper engagement with existing literature may have anticipated these results. Prior studies indicate that the relationship between social media use and mental health is complex, often yielding mixed effects that can be both positive and negative. Moreover, evidence consistently shows that the most vulnerable individuals—such as those with pre-existing mental health challenges—are more likely to experience correlations between intensive social media use and negative mental health outcomes. Incorporating these nuances into your framework would provide a stronger foundation for your analyses.
- Methodological Limitations
- Non-representative and Biased Sampling:
One key limitation is the representativeness of your sample. While including both public and private schools is commendable, the selection process does not appear to have been conducted through probabilistic sampling. By classifying schools based on county size, you included small urban schools and large rural schools. However, the sample selection does not start from a complete listing of schools encompassing the full population of rural and urban schools that constituted the study universe, introducing selection bias, further amplified by self-selection bias (e.g., the study does not report how many schools declined participation and their characteristics). These issues limit the generalizability of your findings. - Insufficient Subsample of LGBT+ Youth:
While your inclusion of LGBT+ youth is a valuable contribution, the subsample size is too small to yield robust and statistically significant conclusions. The small number of LGBT+ participants introduces sensitivity to random variations and undermines the validity of the findings for this group. In future iterations or studies, a larger sample of LGBT+ youth is recommended, or weighting could be used to adjust for representativeness while ensuring general sample analyses remain valid. - Categorical Measures for Social Media Use:
Using categorical measures to assess social media frequency reduces the precision of your findings. This approach may fail to capture subtle variations in usage patterns among participants. - Conclusion
While your study raises valuable questions about the relationship between social media, online social support, and mental health, its conceptual framework and methodological design present significant limitations that may affect your findings. Addressing these issues in future work—such as ensuring representativeness, mitigating biases in school recruitment, and including a larger and more robust subsample of LGBT+ youth—will enable your research to contribute more effectively to this critical area of study.
I appreciate the effort and importance of your work and encourage you to consider these recommendations as an opportunity to refine and enhance the impact of your study. I look forward to seeing how this promising research evolves in subsequent iterations.
Best regards,
Author Response
We would like to thank the reviewers for your thoughtful and helpful feedback on our manuscript titled “The role of online social support in mental health: Comparing rural and urban youth”. We appreciated the positive comments on this manuscript and were grateful for the opportunity to revise and resubmit our work. We agree that investigations into this topic, particularly among such vulnerable populations, are both timely and important. Additionally, we believe the reviewers’ constructive insights will substantially strengthen this manuscript, and we endeavored to address all the reviewers’ comments.
Changes made to the manuscript are highlighted. Below, we have provided our responses to the reviewers’ comments.
Comment 1: Thank you for your submission of the manuscript titled "The role of online social support in mental health: Comparing rural and urban youth." Your study addresses an important and timely topic, exploring the relationship between social media use, online social support, and mental health outcomes among adolescents from rural and urban communities, while also investigating the experiences of LGBT+ youth. However, upon review, I identified several conceptual and methodological concerns that I believe could be addressed to strengthen the impact and clarity of your work.
Response 1: Thank you for the positive comments regarding the important constructs and populations investigated in our study. We have edited many sections throughout the manuscript to address your concerns and believe this has greatly strengthened our work. We hope you will also find that these revisions have improved our manuscript.
Comment 2: Challenges to the Study’s Framework
Your hypothesis that a higher frequency of social media use leads to greater online social support, and consequently to reduced depression and anxiety symptoms, presents an interesting starting point. However, the results of your study suggest that this hypothesis does not hold, as no significant associations were found between online social support and symptoms of depression or anxiety. This challenges the initial premise of your work.
A deeper engagement with existing literature may have anticipated these results. Prior studies indicate that the relationship between social media use and mental health is complex, often yielding mixed effects that can be both positive and negative. Moreover, evidence consistently shows that the most vulnerable individuals—such as those with pre-existing mental health challenges—are more likely to experience correlations between intensive social media use and negative mental health outcomes. Incorporating these nuances into your framework would provide a stronger foundation for your analyses.
Response 2: Thank you for your thoughtful feedback and for drawing our attention to this important area of expansion. We agree that the relationship between social media use and mental health is complex, with mixed effects that can be both positive and negative. We also agree that vulnerable individuals are at greater risk for more negative outcomes online. Additionally, some studies have also found that vulnerable groups report benefitting from their online interactions (De Nardi et al., 2020; Donovan et al., 2021; Nolan et al., 2017; Selkie et al., 2020) and endorse stronger feelings of connectedness. We expanded much of the introduction to provide greater detail about each of these points (pp. 3 and 4) to add clarification and provide a stronger foundation for the purpose of this study. We hope you will agree that this is now improved and supports the foundation of the manuscript as a whole.
Comment 3: Methodological Limitations
Non-representative and Biased Sampling:
One key limitation is the representativeness of your sample. While including both public and private schools is commendable, the selection process does not appear to have been conducted through probabilistic sampling. By classifying schools based on county size, you included small urban schools and large rural schools. However, the sample selection does not start from a complete listing of schools encompassing the full population of rural and urban schools that constituted the study universe, introducing selection bias, further amplified by self-selection bias (e.g., the study does not report how many schools declined participation and their characteristics). These issues limit the generalizability of your findings.
Response 3: Thank you for providing us with the opportunity to ensure we included adequate discussion into the shortcomings of our work. We agree that it would have been ideal to recruit from a complete listing of schools that were within rural and urban counties within our geographical region. We did our best to recruit schools from the broadest range possible. We began recruitment by contacting all schools within a two-hour radius of our University. We eventually expanded this to include schools within a five-hour radius. We contacted approximately 80 schools in total. Many schools declined to participate due to a range of reasons including being in transitional periods (i.e., new school administrators or navigating new resources), having limited time in the school day or after the school day to dedicate to research, and school policy restrictions that did not allow research to be conducted. We have added additional information about these procedures on pg. 6 of the manuscript to add transparency. We have also added greater detail to this section of our limitations on pg. 17.
Comment 4: Insufficient Subsample of LGBT+ Youth:
While your inclusion of LGBT+ youth is a valuable contribution, the subsample size is too small to yield robust and statistically significant conclusions. The small number of LGBT+ participants introduces sensitivity to random variations and undermines the validity of the findings for this group. In future iterations or studies, a larger sample of LGBT+ youth is recommended, or weighting could be used to adjust for representativeness while ensuring general sample analyses remain valid.
Response 4: Thank you for identifying this important consideration. We have added an additional paragraph about this to pg. 17 to highlight this limitation and provide justification for its inclusion in the present manuscript.
Comment 5: Categorical Measures for Social Media Use:
Using categorical measures to assess social media frequency reduces the precision of your findings. This approach may fail to capture subtle variations in usage patterns among participants.
Response 5: Thank you for providing us with the opportunity to discuss this limitation in greater depth. We agree that categorical self-report measures of social media use may not be the first line method for collecting data on social media use frequency. However, the items used in our study were part of a larger measure, the Online Social Support Scale (Nick et al., 2018). The Online Social Support Scale is an established measure, therefore, using this measure will allow the present results to be compared other studies that utilize this same tool. We have added additional details on pp. 17 and 18 to discuss the limitations of self-reported social media use frequency generally and provide suggestions for improved methodology in future research.
Comment 6: Conclusion
While your study raises valuable questions about the relationship between social media, online social support, and mental health, its conceptual framework and methodological design present significant limitations that may affect your findings. Addressing these issues in future work—such as ensuring representativeness, mitigating biases in school recruitment, and including a larger and more robust subsample of LGBT+ youth—will enable your research to contribute more effectively to this critical area of study.
Response 6: Thank you for your kind feedback on the value of this research and your insight into how to enhance this work in future studies. We agree that these limitations may impact the generalizability of the present findings and look forward to strengthening these aspects of our work in the future. We have added details regarding these limitations on pp. 17 and 18.
Comment 7: I appreciate the effort and importance of your work and encourage you to consider these recommendations as an opportunity to refine and enhance the impact of your study. I look forward to seeing how this promising research evolves in subsequent iterations.
Response 7: Thank you very much for your time reviewing our manuscript and your supportive feedback! We have incorporated your feedback accordingly and hope you find that the revisions do indeed refine and enhance this manuscript.
In conclusion, my co-author and I would like to thank the reviewers for their helpful insights and thoughtful attention given to our manuscript. We appreciate the reviewers working with us to strengthen the manuscript. We believe the revised manuscript is a significant improvement on our original submission and we hope that the reviewers agree!
Reviewer 2 Report
Comments and Suggestions for Authors
Thank you for the opportunity to review the manuscript “The role of online social support in mental health: Comparing rural and urban youth”. Overall, I found the writing of this paper is clear and straightforward. However, I have concerns and some suggestions in its current version, especially the rationale of the paper together with the statistical strategy, which I outlined below.
1. I suggest using subheadings in Introduction session to reorganize and clarify the main components, concepts, and research questions.
2. It is not clear why authors anticipate rural youth to report higher levels of social media and social support than urban youth based on information provided in the Introduction. For instance, the authors described different risky factors of urban and rural adolescents, and vulnerable groups from rural settings. Nonetheless, little is available regarding how rural and urban youth may differ in their use of social media, perception of peer support and the relevant reasons.
3. Similarly, it is not clear why the authors anticipated the relation between online support and mental health outcomes would be stronger among rural youths. The authors touched some points throughout the Introduction about rural adolescents may have fewer resources and accessibility to needed health care but the paper lacks a concrete, strong argument to support this hypothesis. Again, I think the paper will be improved through reorganizing the Introduction.
4. The description of differences between rural vs. urban participants should go under “Descriptive Statistics” vs. Methods.
5. Per APA guidelines please use participants instead of sample across the manuscript.
6. If the participants were recruited from a few schools, did the authors consider any clustered feature in analysis? For example, students from the same school may share some levels of peer support and this may be a covariate in your analysis.
7. Depression is measured on a 4-point scale but the item score ranges from 0 to 4, which is 5-point. Please clarify. In addition, it would be helpful to provide the corresponding numeric scores of the lowest and highest response option across surveys to ease understanding.
8. The authors seem to exclude responses with incomplete responses, which is casewise deletion. Such strategy can yield biased results and I don’t recommend handling missing data this way, unless the authors can provide a strong justification of their rationale. The authors also employed mean imputation to handle missing data, which is problematic as single imputation will generate biased results.
9. If the authors anticipated differences from different aspects of social support, the concepts of these subscales need to be discussed upfront in the Introduction and relevant theoretical justification is needed. It is also confusing why the authors used subtypes of social support in some but not all analyses. The current writing of Introduction only discusses overall social support but the actual analyses and Discussion include subtypes of social support, generating a content disconnection of the manuscript.
10. It was not clear until the Results session how the authors collected social media use data. These information needs to be presented in the Methods section. Similarly, the different types of social media and how they can be related to other key concepts of this paper were not discussed until the Discussion section. If authors anticipate different findings based on different types of social media, they should be discussed earlier in the manuscript.
11. Some wordings in the Discussion need carefully attention. For instance, “In the current study, however, social media use was not found to be associated with anxiety or depression among adolescents from rural or urban community types. This lack of an association between social media use and psychopathology is in line with some more recent studies which have suggested that social media use, when used appropriately, may aid in promoting social connectedness and strengthening relationships.” The lack of association does not necessarily indicate a positive role of social media use in social relationships. Likewise, “An explanation for these findings may be that while online social support may be beneficial under stress, depression and anxiety are fundamentally different from stress, and thus this type of online social support may not function in the same way when these pathologies are present.” It is not clear what the authors are arguing here.
12. I may not understand the argument correctly. The authors stated the use of paper-and-pencil survey as a strength of this paper from a methodological perspective. Although paper-and-pencil survey allows more accessibility to participation, it is not clear how this is relevant to this current study, especially when most participants should have some extent of Internet use experiences. In addition, the concerns of “bot” responses or inattentive responses sound more like a concern when the survey is distributed through more public channels (e.g., when participants are recruited online universally through online ed and no interaction is needed between the researcher and participant), but this type of concerns seems to be less worrisome in the current situation. In contrast, paper-and-pencil survey may introduce more human error in data entry processes. I recommend the authors to reconsider this part more consciously
13. Please check the spelling throughout, there are incorrect spelling of “LGTQ+” and missing letters of Instrumental support on p.12.
Author Response
We would like to thank the reviewers for your thoughtful and helpful feedback on our manuscript titled “The role of online social support in mental health: Comparing rural and urban youth”. We appreciated the positive comments on this manuscript and were grateful for the opportunity to revise and resubmit our work. We agree that investigations into this topic, particularly among such vulnerable populations, are both timely and important. Additionally, we believe the reviewers’ constructive insights will substantially strengthen this manuscript, and we endeavored to address all the reviewers’ comments.
Changes made to the manuscript are highlighted. Below, we have provided our responses to the reviewers’ comments.
Comment 1: Thank you for the opportunity to review the manuscript “The role of online social support in mental health: Comparing rural and urban youth”. Overall, I found the writing of this paper is clear and straightforward. However, I have concerns and some suggestions in its current version, especially the rationale of the paper together with the statistical strategy, which I outlined below.
Response 1: Thank you for your positive comments regarding the writing of our manuscript. We appreciate your helpful suggestions for how to improve and strengthen the impact of our study. We hope you find that our revisions have resulted in a stronger manuscript overall and we look forward to your feedback on this next iteration.
Comment 2: I suggest using subheadings in Introduction session to reorganize and clarify the main components, concepts, and research questions.
Response 2: Thank you for this suggestion. We have added subheadings throughout the introduction on pp. 1-4 to provide greater clarity and structure.
Comment 3: It is not clear why authors anticipate rural youth to report higher levels of social media and social support than urban youth based on information provided in the Introduction. For instance, the authors described different risky factors of urban and rural adolescents, and vulnerable groups from rural settings. Nonetheless, little is available regarding how rural and urban youth may differ in their use of social media, perception of peer support and the relevant reasons.
Response 3: Thank you for identifying this important omission. Rural youths are known to benefit greatly from a strong social support network (Newland et al., 2014). They have also been identified as among the most frequent social media users (McInroy et al., 2018). While the research on associations between social media use and online social support among rural youth is very slim, research on other vulnerable or isolated populations has demonstrated that social media can be an accessible means of garnering support from others outside of one’s proximal community (e.g., Karim et al., 2022, Escobar-Viera et al., 2022; Selkie et al., 2020). In fact, two studies of LGBT+ youth in rural communities found that these youth reported several benefits of social media use including improved informational and emotional support (Karim et al., 2022, Escobar-Viera et al., 2022). It is likely that other youth in rural communities use social media in similar ways, as social media use has been found to augment perceptions of community connectedness among rural adolescents (Hampton & Shin., 2023). We have expanded the introduction and added greater detail regarding the evidence base and rationale for this hypothesis on pg. 4 of the introduction. We hope that you will find this better supports our hypotheses and improves the organization of the manuscript as a whole.
Comment 4: Similarly, it is not clear why the authors anticipated the relation between online support and mental health outcomes would be stronger among rural youths. The authors touched some points throughout the Introduction about rural adolescents may have fewer resources and accessibility to needed health care but the paper lacks a concrete, strong argument to support this hypothesis. Again, I think the paper will be improved through reorganizing the Introduction.
Response 4: Thank you again for the opportunity to clarify our rationale. We have reorganized our introduction and added more subheadings to add clarity and organization to our introduction. We have also added greater depth on pp. 3 and 4 regarding how social media use may contribute to perceptions of online social support among rural youth. We also added a few clarifying sentences throughout the introduction and reorganized subsection 1.2 on pg. 2 to emphasize the importance of social support regarding psychological health within rural communities. We hope you find that these additions and organizational changes strengthen our arguments and the foundation for our hypotheses.
Comment 5: The description of differences between rural vs. urban participants should go under “Descriptive Statistics” vs. Methods.
Response 5: Thank you for pointing out this organizational error. The rural vs. urban group differences have been moved to the Descriptive Statistics section on pg. 8.
Comment 6: Per APA guidelines please use participants instead of sample across the manuscript.
Response 6: Thank you for this correction. “Sample” was changed to “participants” throughout the manuscript.
Comment 7: If the participants were recruited from a few schools, did the authors consider any clustered feature in analysis? For example, students from the same school may share some levels of peer support and this may be a covariate in your analysis.
Response 7: Thank you for this helpful question. We agree that it would be interesting and important to evaluate whether schools were covariates in our study outcomes. However, the student body sizes of the participating schools were highly discrepant. That is participant numbers for each school were 7, 8, 18, 25, 33, 73, and 111. Given this broad range, it would not be possible to effectively identify whether schools were covariates in the present analyses. We will keep this suggestion in mind for future studies as we agree this is an important factor to consider. We have added this as a limitation and future direction on pg. 17.
Comment 8: Depression is measured on a 4-point scale but the item score ranges from 0 to 4, which is 5-point. Please clarify. In addition, it would be helpful to provide the corresponding numeric scores of the lowest and highest response option across surveys to ease understanding.
Response 8: Thank you for pointing out this error. We have corrected this and added the corresponding numeric scores and response options for all measures listed on pp. 6 and 7.
Comment 9: The authors seem to exclude responses with incomplete responses, which is casewise deletion. Such strategy can yield biased results and I don’t recommend handling missing data this way, unless the authors can provide a strong justification of their rationale. The authors also employed mean imputation to handle missing data, which is problematic as single imputation will generate biased results.
Response 9: Thank you for bringing up this point. We agree that more sophisticated methods, such as multiple imputation, are the preferable method when there is >5% of missing data. Our study was fortunate to have very little missing data as we only had 0.003% missingness (66 out of 18,961 data points). Specifically, five participants missed 1 item on the GAD-7, eleven participants missed 1 item and three participants missed 2 on CESD, and thirteen participants missed 1 item on the OSSS while nine participants missed between 2 and 7 out of the possible 40 items on this measure. Given this very low amount of missing data, single imputation is an appropriate method (Madley-Dowd, Hughes, Tilling, Heron, 2019). We have added this information to the manuscript on pg. 7. Thank you for pointing out this necessary clarification.
Comment 10: If the authors anticipated differences from different aspects of social support, the concepts of these subscales need to be discussed upfront in the Introduction and relevant theoretical justification is needed. It is also confusing why the authors used subtypes of social support in some but not all analyses. The current writing of Introduction only discusses overall social support but the actual analyses and Discussion include subtypes of social support, generating a content disconnection of the manuscript.
Response 10: Thank you for bringing our attention to this disconnect between the various sections of our manuscript. We have added further information on pp. 3 and 4 on the literature base and theoretical underpinnings behind examining online social support subtypes. We hope that this addition aids in better connecting the content across the entire manuscript, adding clarity and continuity. Additionally, we thank you for highlighting our omission of including subscales of social support across all analyses. We have added additional statistics to pp. 11-13 to ensure the subscales of social support were evaluated in all analyses.
These additional findings were largely insignificant. More specifically, in the depression models among all youth, there were no main effects found for esteem/emotional support (β = -0.050, SE = 0.101, p = .526), social companionship (β = 0.021, SE = 0.095, p = .795), informational support (β = 0.028, SE = 0.106, p = .731), or instrumental support (β = -0.011, SE = 0.110, p = .890). Similarly, in the anxiety models among all youth, there were no main effects found for esteem/emotional support (β = 0.044, SE = 0.041, p = .557), social companionship (β = 0.067, SE = 0.038, p = .378), informational support (β = 0.037, SE = 0.043, p = .637), or instrumental support (β = -0.017, SE = 0.045, p = .821).
Similar results were demonstrated among LGBT+ youth. That is, in the depression models, there were no significant main effects for esteem/emotional support (β = -0.024, SE = 0.211, p = .888), social companionship (β = -0.083, SE = 0.210, p = .640), or informational support (β = 0.164, SE = 0.245, p = .385). In the anxiety models, there were also no main effects for esteem/emotional support (β = 0.110, SE = 0.079, p = .514), social companionship (β = 0.121, SE = 0.078, p = .488), or informational support (β = 0.376, SE = 0.089, p = .035). However, instrumental support was found to have a significant positive main effect on both de-pression (β = 0.409, SE = 0.301, p = .046) and anxiety (β = 0.453, SE = 0.11, p = .023) such that instrumental support was higher among youth with greater symptom severities. We include these new results on pp. 11-13 and provide a brief interpretation of the significant findings on pg. 17.
Comment 11: It was not clear until the Results session how the authors collected social media use data. These information needs to be presented in the Methods section. Similarly, the different types of social media and how they can be related to other key concepts of this paper were not discussed until the Discussion section. If authors anticipate different findings based on different types of social media, they should be discussed earlier in the manuscript.
Response 11: Thank you for providing us with the opportunity to further elaborate on this important component of our manuscript. We added clarifying details to the methods section on pg. 6 regarding how social media use frequency data were collected. We also added additional background information on pp. 3 and 4 regarding research on benefits associated with different types of social media. We discuss how different types of online engagement are associated with various benefits and more clearly highlight the gap in the literature regarding investigations into social media platforms and specific types of online social support. We hope you will find that this improved the continuity throughout our manuscript.
Comment 12: Some wordings in the Discussion need carefully attention. For instance, “In the current study, however, social media use was not found to be associated with anxiety or depression among adolescents from rural or urban community types. This lack of an association between social media use and psychopathology is in line with some more recent studies which have suggested that social media use, when used appropriately, may aid in promoting social connectedness and strengthening relationships.” The lack of association does not necessarily indicate a positive role of social media use in social relationships. Likewise, “An explanation for these findings may be that while online social support may be beneficial under stress, depression and anxiety are fundamentally different from stress, and thus this type of online social support may not function in the same way when these pathologies are present.” It is not clear what the authors are arguing here.
Response 12: Thank you for drawing our attention to this issue and allowing us to resolve the language in these statements. Regarding the first statement, we have softened some language throughout the discussion section on pp. 13-16 as we agree that the previous statement was not necessarily warranted by our present findings. We have also expanded our explanation on pg. 16 regarding your second point to better clarify our argument. We hope you will find that these add clarity and strength to these conclusions.
Comment 13: I may not understand the argument correctly. The authors stated the use of paper-and-pencil survey as a strength of this paper from a methodological perspective. Although paper-and-pencil survey allows more accessibility to participation, it is not clear how this is relevant to this current study, especially when most participants should have some extent of Internet use experiences. In addition, the concerns of “bot” responses or inattentive responses sound more like a concern when the survey is distributed through more public channels (e.g., when participants are recruited online universally through online ed and no interaction is needed between the researcher and participant), but this type of concerns seems to be less worrisome in the current situation. In contrast, paper-and-pencil survey may introduce more human error in data entry processes. I recommend the authors to reconsider this part more consciously
Response 13: Thank you for allowing us this opportunity to clarify. The strengths of each methodological approach (paper-pencil and online) are complex and are both accompanied by unique potential shortcomings. We agree it was important for participants to have at least some engagement with social media platforms to participate in this study. We addressed this concern preemptively during the recruitment phase of the study such that, when participants were provided with information about the study, they were also informed that at least some phone/social media use was required for participation. We have added details about this on pg. 6. We further confirmed this upon reviewing our data and found no participants to report a total absence of social media/phone use. If we had used online recruitment methods, we may have been at greater risk for only recruiting frequent social media users, as they would have been more likely to be exposed to advertisements for the study. Furthermore, during the school recruitment phase, school principals stated their students would not be able to complete the survey online during school hours due to limitations on phone use in some school policies. Thus, the paper-and-pencil method allowed us to follow school policies and procedures. We have added additional information to elaborate on this on pg. 17. We also removed the statements about safeguarding against “bot” responses. As for the concern about paper-and-pencil methods introducing more human error in data entry processes, we used a protocol in which two separate research assistants were responsible for entering the same data and the data entry was checked for inconsistencies – if there was an inconsistency across the two research assistants, the data was reviewed by the principal investigator to ensure accuracy. We believe that this protocol effectively mitigated data entry errors.
Comment 14: Please check the spelling throughout, there are incorrect spelling of “LGTQ+” and missing letters of Instrumental support on p.12.
Response 14: Thank you for pointing out these errors. We have corrected them and have closely reviewed the remainder of the manuscript to identify any additional spelling errors.
In conclusion, my co-author and I would like to thank the reviewers for their helpful insights and thoughtful attention given to our manuscript. We appreciate the reviewers working with us to strengthen the manuscript. We believe the revised manuscript is a significant improvement on our original submission and we hope that the reviewers agree!
Round 2
Reviewer 1 Report
Comments and Suggestions for Authors
Dear Authors,
Thank you for your thoughtful responses to the feedback provided during the review process for your manuscript, “The role of online social support in mental health: Comparing rural and urban youth.” I appreciate the effort you have dedicated to addressing the comments and suggestions, which have significantly strengthened your work.
I am particularly impressed by the notable improvements in the background and theoretical framework of the study. The expanded introduction provides a clearer and more comprehensive rationale for the research, integrating relevant literature and contextual nuances. This not only reinforces the importance of your study but also situates it more effectively within the broader field of research on social media use, mental health, and vulnerable populations such as rural and LGBT+ youth.
Additionally, I commend your acknowledgment of the study's limitations. By addressing methodological concerns, such as sample representativeness, recruitment challenges, and the small LGBT+ subsample size, you have enhanced the rigor and transparency of your work. The detailed discussions on these points in the revised manuscript contribute to its credibility and provide a solid foundation for future research.
Your responses reflect a commitment to improving the manuscript and advancing knowledge in this critical area of study. The revisions have resulted in a more robust piece of research. I look forward to seeing the contributions this work will make to the field.
Thank you again for your dedication and for engaging constructively with the feedback provided. It has been a pleasure to review your manuscript.
Best wishes,